# Estimating Black Oat Biomass Using Digital Surface Models and a Vegetation Index Derived from RGB-Based Aerial Images

Lucas Renato Trevisan [1],*, Lisiane Brichi [1], Tamara Maria Gomes [1,2] and Fabrício Rossi [1,2]

1 "Luiz de Queiroz" College of Agriculture, University of São Paulo, 11 Pádua Dias Avenue, São Paulo 13418-900, Brazil
2 Faculty of Food Engineering and Animal Sciences, University of São Paulo, 225 Duque de Caxias Avenue, São Paulo 13635-900, Brazil
* Correspondence: lucas.renato.trevisan@usp.br; Tel.: +55-14-98192-2925

**Abstract:** Responsible for food production and industry inputs, agriculture needs to adapt to worldwide increasing demands and environmental requirements. In this scenario, black oat has gained environmental and economic importance since it can be used in no-tillage systems, green manure, or animal feed supplementation. Despite its importance, few studies have been conducted to introduce more accurate and technological applications. Plant height (H) correlates with biomass production, which is related to yield. Similarly, productivity status can be estimated from vegetation indices (VIs). The use of unmanned aerial vehicles (UAV) for imaging enables greater spatial and temporal resolutions from which to derive information such as H and VI. However, faster and more accurate methodologies are necessary for the application of this technology. This study intended to obtain high-quality digital surface models (DSMs) and orthoimages from UAV-based RGB images via a direct-to-process means; that is, without the use of ground control points or image pre-processing. DSMs and orthoimages were used to derive H ($H_{DSM}$) and VIs ($VI_{RGB}$), which were used for H and dry biomass (DB) modeling. Results showed that $H_{DSM}$ presented a strong correlation with actual plant height ($H_{REF}$) ($R^2 = 0.85$). Modeling biomass based on $H_{DSM}$ demonstrated better performance for data collected up until and including the grain filling ($R^2 = 0.84$) and flowering ($R^2 = 0.82$) stages. Biomass modeling based on $VI_{RGB}$ performed better for data collected up until and including the booting stage ($R^2 = 0.80$). The best results for biomass estimation were obtained by combining $H_{DSM}$ and $VI_{RGB}$, with data collected up until and including the grain filling stage ($R^2 = 0.86$). Therefore, the presented methodology has permitted the generation of trustworthy models for estimating the H and DB of black oats.

**Keywords:** image analysis; plant growth; computer vision; remote sensing; visible spectrum; unmanned aerial vehicle; plant height

## 1. Introduction

Alongside the fast pace of technological development, society has been increasing its demand for food and supplies. Moreover, environmental requirements exert pressure on the development of more sustainable processes. Agriculture is implicated in this context, being responsible for food production and many industry inputs, which emphasizes the need for more sustainable forms of production. Methods such as the no-tillage system and the use of green manure have gained prominence over recent years since they contribute to more sustainable agricultural processes, reduce risks and environmental impacts, recycle nutrients, and increase organic matter in the soil [1].

In this context, black oat has gained prominence for being one of the most important winter grasses around the world that can be used as a cover crop [2] or for supplementation in animal feed [3,4]. Furthermore, its application as green manure has great potential; it is often applied in crop succession systems and is able to bring chemical and physical benefits

to the soil and increase yield for the subsequent crop [5]. In addition, black oat can also be successfully applied to reduce weed infestations [6]. Despite their importance, black oat and other cover crops are rarely studied concerning new technologies, presenting a research gap in this area.

Plant biomass is directly related to development and yield. Studies have shown that black oat height correlates with biomass production and nitrogen ratios; thus, prediction models can be built based on this parameter [7,8]. Similarly, productivity and nutritional status can be estimated from vegetative indices determined through images [7,9]. It can be understood that nitrogen and biomass, as well as plant height, share a strong correlation.

In the related literature, many studies regarding biomass and nitrogen status for crops such as rice, corn, and wheat (Table 1) can be found. It is important to highlight that the most important cereals, including corn, wheat, rice, and oats, belong to the grass family, which leads to a high correlation in their physiological behavior [10]. Consequently, it is possible to relate the results obtained through $VI_{RGB}$ and plant height applied, for example, to corn, wheat, and rice when assessing biomass and nitrogen status, with the results obtained through the same $VI_{RGBS}$ and plant height applied to black oat.

Therefore, studying the physiological and morphological aspects of black oat becomes interesting because they present correlations with response variables of interest, such as biomass [11]. Nonetheless, methodologies for determining plant biomass are usually destructive, damaging the crop site and demanding extensive labor, thus leading to a need for alternatives that could provide biomass assessment in faster and non-destructive forms. A very advantageous alternative is the determination of $VI_{RGB}$ and plant height through the use of aerial images, which can provide information related to productivity since they are directly related to biomass [12].

In the context of aerial imaging, the use of unmanned aerial vehicles (UAV) has gained prominence [13]. When compared with satellite images, the use of AUV presents advantages. One of the most prominent factors in its application is the possibility of having high spatial and temporal resolution [13]. UAVs present greater flexibility regarding the sample distance (flight height), mission time, and the ease of mission repetition [14]. Another important advantage of UAV-based imaging is the gain in especial and temporal resolution that allows one to obtain information with a high level of detail over time [15], especially due to the ease of performing flights with greater frequency and the possibility of flying at lower heights, which is known as low-altitude remote sensing (LARS) [16].

A UAV can be equipped with several different types of sensors, such as cameras that operate in the visible, multispectral, and hyperspectral ranges of the electromagnetic spectrum [17]. However, the use of multi- and hyper-spectral cameras requires greater and more complex processing to extract information, making their use unfeasible in many applications, which justifies the use of images in the visible spectrum [18].

The visible band of the spectrum is contained within wavelengths ranging from approximately 400 nm to 700 nm [19]. A spectral signature of a plant leaf contains important information. Between spectral wavelengths of 500 and 550 nm, there is a peak of reflectance that characterizes a response region for photosynthetic efficiency. It is noteworthy that the visible spectrum is within this range, which ensures the possibility of extracting spectral information for plant analysis [20].

At wavelengths of approximately 500 nm, 600 nm, and 700 nm, a region of responses to chlorophyll and carotenoids is present; this is an important region for studying nutrient contents and phenology. The ramp between approximately 700 nm and 800 nm is the region known as the red edge because this is where infrared begins; this is an important region for the evaluation of green biomass [20].

Therefore, reflectance response is observed within the wavelengths of the visible spectrum, justifying applications in this region. Visible spectrum aerial images have been successfully applied in many applications such as the determination of gaps in grape yards [21], and the development of prediction models for sugarcane using visible-band vegetation indices [22]. Acorsi et al. [7] studied prediction models for black oat productivity

using RGB-based UAV images and are one of very few studies involving this crop and technology.

Furthermore, the use of images obtained by UAV coupled with RGB cameras is a viable alternative for monitoring biomass [23]. This approach has been applied with success to crops such as barley [12], cotton [24], wheat [25], black oat [7], corn [26], soybean [27], and potato [28]. Though they can be found in the literature, applications to black oat are very scarce.

Besides plant height, the productive aspects of plants can be assessed through vegetation indices (VI), which are used to extract information from crops that allow for the derivation of classification models [29]. An extended review on the use of reflectance techniques stated that photosynthetic pigments present in leaves could be used to describe the physiological status of a plant as well as to evaluate its biomass [20]. VIs can be used within different ranges of the electromagnetic spectrum, and studies have shown potential applications for VIs that operate in the visible range of the spectrum. RGB-based vegetative indices ($VI_{RGB}$) from UAV-based images can be applied to estimate forage biomass and generate trustworthy results [9]. Table 1 presents some studies from the literature that applied $VI_{RGB}$.

**Table 1.** Vegetation indices from the literature used in this study.

| Name | Abbreviation | Equation | Application | Reference |
|---|---|---|---|---|
| Visible Atmospherically Resistant Index | VARI | $VARI = (G - R)/(G + R + B)$ | Corn biomass | [30] |
| Normalized Green Red Difference Index | NGRDI | $NGRDO = (G - R)/(G + R)$ | Grass biomass | [9] |
| Normalized Excess Green Index | NExG | $NExG = (2G - R - B)/(R + G)$ | Rice nitrogen | [31] |
| Green–Red Ratio Index | GRRI | $GRRI = G/R$ | Corn nitrogen | [32] |
| True Color Vegetation Index | TCVI | $TCVI = (2R - 2B)/(2R - G - 2B + 255 \times 0.4)$ | Wheat nitrogen | [33] |

Methods for determining biomass that are fast and practical are needed, since in many situations in agriculture, manual methods are time-consuming, depend on extensive training and labor experience, and decision making can vary according to each person performing the tasks [34]. Digital surface models (DSMs) are widely used in agricultural applications, such as crop monitoring for vegetation analysis [21], because they allow the estimation of plant height which, in turn, is related to biomass. The current availability of software based on structure from motion (SfM) techniques facilitates processing for obtaining photogrammetric products such as DSMs and orthoimages. DSMs are models that represent objects at soil level and can be largely used in agricultural applications such as the monitoring of crops [21].

Bendig et al. [35] determined plant height through a DSM ($H_{DSM}$) based on aerial images in the visible spectrum to monitor barley biomass over time, and the study produced satisfactory results. In general, DSMs from UAV images can be applied over time to determine plant height and model plant growth [36] and biomass [11], since it allows high special and temporal resolutions. Moreover, combinatory approaches might increase biomass modeling capabilities. UAV-based aerial images can be used to create biomass prediction models combining $H_{DSM}$ with $VI_{RGB}$, producing better results than models using single factors, such as $VI_{RGB}$ or $H_{DSM}$ alone [11].

When it comes to UAV images, some techniques might impair faster parameter determinations. The use of ground control points (GCP) is important in many applications, even though it requires field work in the form of placing signs and measuring high-precision coordinates, demands specific and expensive equipment, and generates greater computational costs for processing. However, Yue et al. [37] published a study using DSMs for

productivity estimation and demonstrated that $H_{DSM}$ presented a high correlation with biomass even without using GCPs, thus demonstrating that for fast and non-destructive yield estimation, the application of GCPs might not be essential [26]. It is important to mention that for situations that require a greater precision of coordinates, such as variable rate fertilizer applications, GCPs are indispensable.

Therefore, this study intended to obtain high-quality DSMs and orthoimages from UAV-based RGB images via direct-to-process means; that is, without the use of ground control points or image pre-processing, such as light correction or color calibration, thus offering a faster and more accurate methodology for obtaining these photogrammetric products. Additionally, the present study aimed to estimate black oat dry biomass using regression models based on $H_{DSM}$ and $VI_{RGB}$ alone as well as the combination of $H_{DSM}$ and $VI_{RGB}$.

## 2. Materials and Methods

The graphical abstract presented in Figure 1 concisely represents the experimental workflow used in this study. A UAV coupled with a camera operating in the visible band of the spectrum was used to acquire RGB images from black oat crops exposed to natural light. The flight missions for image acquisition were always performed under similar conditions, i.e., clear sky, no wind, and at approximately noon to avoid interference and differences in the images.

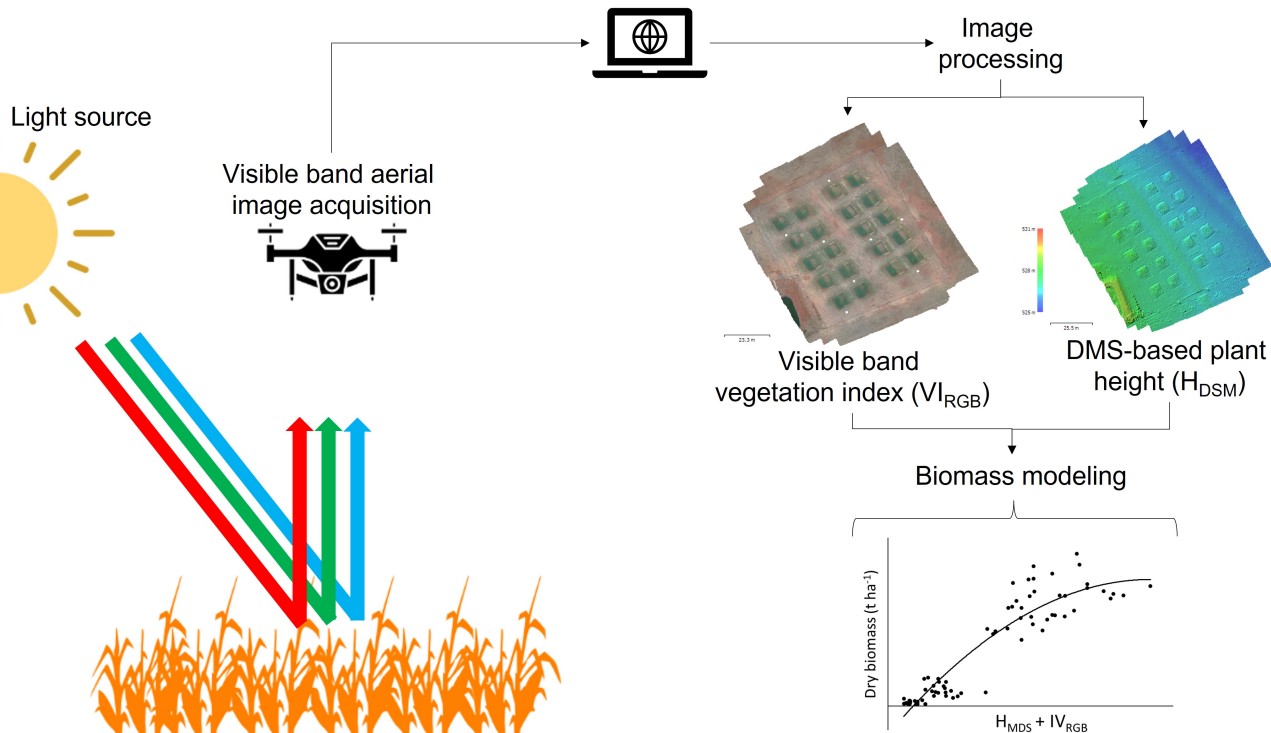

**Figure 1.** Graphical abstract of the present study to estimate black oat biomass using RGB-based UAV images.

Raw images were processed in a computer to obtain the orthoimages and the digital surface models from which visible band vegetation indices ($VI_{RGB}$) and plant height ($H_{DSM}$) were derived, respectively. Later, $VI_{RGB}$ and $H_{DSM}$ were employed for biomass modeling.

### 2.1. Experimental Site

The experimental site is located at the Faculty of Food Engineering and Animal Sciences, University of São Paulo, state of São Paulo, Brazil. The geographical location of the campus is $21°59'$ south latitude and $47°23'$ west longitude and has an average altitude

of 635 m, with declivity. According to the Koppen classification, the climate is Cwa with an average annual temperature and a precipitation of 20.8 °C and 1298 mm, respectively. Finally, the soil at the site is classified as Eutrophic Red Latosol. The experimental site was divided into 20 plots of 8 by 8 m (64 m$^2$), separated into four blocks perpendicular to the direction of the declivity. Alongside each block, a reference area was kept clean to be used as a reference for zero height.

The black oat (cultivar EMBRAPA-29) was sown in the plots within a usable area of 7 by 7 m, in rows 20 cm apart. The design was in randomized blocks with 4 blocks and 5 treatments (T1 to T5); namely, (T1—control) 100% of the recommended dosage for the culture (using nitrogen fertilizer and irrigation with tap water) and (T2) 25%, (T3) 50%, (T4) 75%, and (T5) 100% of treated slaughterhouse effluent (STE) supplied through sprinkler irrigation, which provided different doses of nitrogen. The percentages of STE were combined with tap water to maintain the soil tension between the field capacity and the critical tension.

### 2.2. Image Acquisition and Processing

The image acquisition was performed using a UAV (model Phantom 4 Pro, SZ DJI Technology Co., Shenzhen, China) featuring a digital camera operating in the visible range of the spectrum with 12.4 megapixels (MP) of resolution, 20 mm of focal length, and an angle of view of 94°. The images were collected once a week with autonomous flights performed at approximately noon on clear days to minimize variations influenced by the natural light in the sunlight spectrum. Since the sensor that was coupled with the UAV was only sensible to the visible band of the spectrum, information on non-visible bands, such as infrared, was not captured.

The autonomous inspection flights were configured and controlled by the Pix4D Capture software (Version 4.9.0, Pix4d S.A., Prilly, Switzerland), with lateral and frontal overlap of 70%, and with the camera positioned perpendicularly to the UAV movement direction and 20 m above the ground. The images were processed on a computer (model PE07CAK5, Lenovo Corp., Hong Kong, China) composed of an Intel Core i7 processor (model 8564U, Inter Corp., Santa Clara, CA, USA), 12 GB of RAM, and a GeForce video board with 2 GB of RAM (model MX 110, NVIDIA Inc., Santa Clara, CA, USA).

The orthoimages and DSMs were obtained using the SfM software Agisoft Metashape Professional (Version 1.6.4, Agisoft LLC, St. Petersburg, Russia) in its free trial standard version. The processing workflow included: image alignment, building a dense point cloud, building mesh, building the digital surface model, and building the orthoimage.

In order to present reference measurements that allow for the evaluation of the estimates, in-field sampling was performed for plant height and biomass. For plant height, on the same days the flight missions were performed, in-field reference height measurements ($H_{REF}$) were conducted using a 1.5-m ruler. The measurements were taken randomly at thirty points within each experimental plot, measuring the distance from the ground to the highest point of the plant canopy. The average height was calculated, obtaining one mean value for each plot at each assessment date.

Additionally, biomass samples were collected at each growth stage. To determine dry biomass, a 1 m$^2$ sample was collected from the field, considering the aerial part of the plant. Afterwards, the samples were taken to the laboratory, weighed, cleaned, and set to dry in a forced-air convection oven for 72 h at 65 °C. After, the samples were weighed again to determine dry biomass content.

### 2.3. Processing the Orthoimages and Digital Surface Models

To obtain the $H_{DSM}$ and $VI_{RGB}$ values, the experimental plots for each flight were clipped from the original DSM and orthoimage. All of the processing for image adjustment and clipping of experimental plots was performed using QGIS (Open-Source Geospatial Foundation, version 3.22, Białowieża). To permit the use of one mask for clipping all

images, it was necessary to adjust the coordinates so all the images were aligned over the same spatial points. The adjustment was made using the georeferencer raster in QGIS.

After adjusting the coordinates, the images were aligned and the clipping raster function in QGIS was used to clip each experimental plot from all orthoimages and DSMs, thus generating a total of 20 sub-images from each map and flight date. The resulting clipped sub-images from the DSMs and orthoimages were renamed for easier identification and stored in a cloud drive. An algorithm developed in Python was developed to access those images and extract desired information.

In the first step, the algorithm was programmed to read all clipped images from the DSM and determine and store the mean height values for each reference plot by reading the value of each pixel, computing its sum and dividing by the total number of pixels. Next, the algorithm could open each of the experimental plots and subtract the mean height values from each corresponding experimental plot, and then calculate the mean height value for each plot. This resulted in a mean height value, in meters, with ground level altitude equal to zero. In the end, this resulted in one mean height value for each plot from each flight mission date.

Lastly, the Python algorithm was able to access all sub-images from the orthoimage and calculate the $VI_{RGB}$ values presented in Table 1 (VARI, NGRDI, NExG, GRRI, TCVI). To this end, the channels of each visible band (red, green, and blue) were split, and the VIs were calculated using the intensity of the pixels through point operations. The mean values of each $VI_{RGB}$ for each plot and date were determined, generating one mean IV value for each plot.

*2.4. Statistical Analysis*

Data were analyzed according to the treatments to identify significant differences due to nitrogen dosages through analysis of variance (ANOVA) with a significance level of 0.05. Similarly, the coefficient of determination ($R^2$) and the coefficients of the models were tested for significance ($p < 0.05$) and non-significant coefficients ($p > 0.05$) were discarded. All statistical analysis involved in this study was performed using the RStudio software (RStudio, version 4.2.2, Boston, MA, USA).

Modeling was performed using three datasets considering the growth stages. The first dataset considered data collected up until and including the grain filling stage (all data). The second dataset comprehended the data collected up until and including the flowering stage, thus including data from the tillering, booting, and flowering stages. The third dataset considered the data collected up until and including the booting stage; that is, data from the tillering and booting stages. The effects of the data from single stages alone were not considered since initial studies showed low prediction potential for modeling, with regression models presenting no significance ($p > 0.05$).

To determine the best model for each case of study, the models were fitted to the respective data set, and the significance of the $R^2$ and the coefficients were analyzed. Linear and polynomial single and multiple regression were tested through regression analysis, non-significant models were disregarded, and non-significant coefficients were dropped from the model. From the resulting significant models, the best model with greatest $R^2$ and lowest RMSE was selected for the respective dataset.

To identify the vegetation index that best related to the data obtained in this study, regarding plant height and biomass, Pearson's correlation was performed and the $VI_{RGB}$ with the highest score was selected for modeling.

For the model validations, each one of the three aforementioned datasets was split into different subsets; that is, modeling and validation datasets. To split the data, 70% of the available data in the respective dataset was randomly selected for modeling, and 30% for validation. This approach was applied for both plant height and dry biomass models.

## 3. Results

The UAV flights were conducted for image acquisition, and processing was performed following the methodology workflow previously presented. The flight mission was repeated eight times, including two flights within each development stage of the black oat crop. From the images acquired during each flight, high-resolution DSMs and orthoimages were derived. The performance of these photogrammetric products is depicted in Table 2. Furthermore, Figure 2 presents the resulting DSM and orthoimage for one of the dates when a flight mission occurred, to demonstrate the photogrammetric products obtained.

**Table 2.** Descriptive information from photogrammetric products.

| Flight | Date [1] | DAS [2] | Growth Stage | Density (pt. m$^{-2}$) [3] | Image Overlap [4] | DSM Resolution (mm pix$^{-1}$) | Orthoimage Resolution (cm pix$^{-1}$) |
|---|---|---|---|---|---|---|---|
| 1 | 10 August 2020 | 11 | Tillering | 6941.72 | >8 | 6.29 | 1.26 |
| 2 | 18 August 2020 | 19 | Tillering | 7139.54 | >8 | 6.22 | 1.24 |
| 3 | 25 August 2020 | 26 | Booting | 6977.67 | >8 | 6.25 | 1.25 |
| 4 | 1 September 2020 | 33 | Booting | 7191.29 | >8 | 6.06 | 1.21 |
| 5 | 8 September 2020 | 40 | Flowering | 7234.58 | >8 | 6.21 | 1.24 |
| 6 | 15 September 2020 | 47 | Flowering | 7107.64 | >8 | 6.20 | 1.24 |
| 7 | 23 September 2020 | 55 | Grain filling | 7549.89 | >8 | 6.13 | 1.23 |
| 8 | 30 September 2020 | 62 | Grain filling | 7277.06 | >8 | 6.28 | 1.26 |

[1] Every date occurred in 2020. [2] Days after sowing. [3] Point density in the dense point cloud. [4] Number of images over a common point.

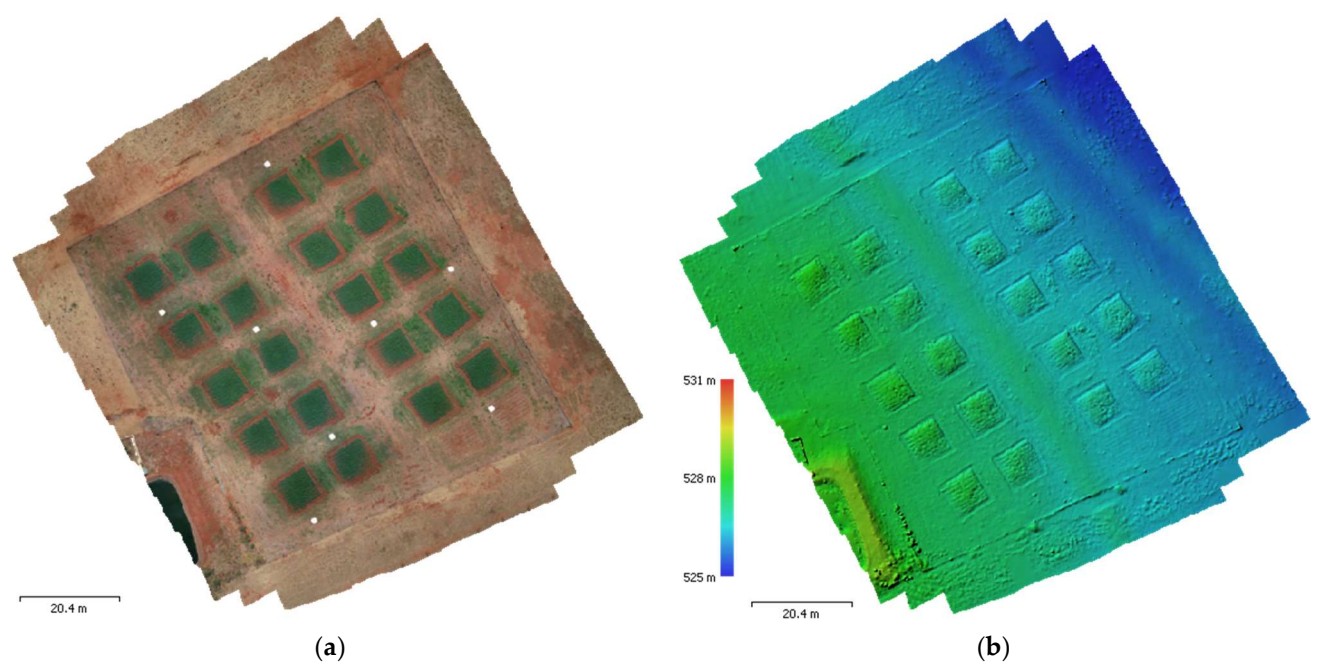

(**a**)　　　　　　　　　　　　　　　　　　　　　(**b**)

**Figure 2.** Photogrammetric products resulting from the flight performed on September 15. (**a**) Orthoimage. (**b**) Digital surface model.

### 3.1. Plant Height Assessment

Plant height was derived from the DSM ($H_{DSM}$) through image processing. Correspondingly to each UAV flight, ground references of plant height ($H_{REF}$) were obtained via in-field measurements. Table 3 presents the descriptive statistics of $H_{DSM}$ and $H_{REF}$ values including mean, standard deviation, and height range. When considering the overall mean from all plant heights of the respective flight date, a standard deviation ranging from 0.01 to 0.17 m was observed for the $H_{REF}$ measurements, while the $H_{DSM}$ measurements generated lower standard deviations, which ranged from 0.01 to 0.07 m. These lower

standard deviations might be related to the greater number of points in the determination of $H_{DSM}$ through the aerial images.

**Table 3.** Descriptive statistics of $H_{DSM}$ and $H_{REF}$ for each flight mission and in-field inspection.

| Flight | $H_{REF}$ (m) | | | | | $H_{DSM}$ (m) | | | | |
|---|---|---|---|---|---|---|---|---|---|---|
| | μ | σ | Min | Max | Range | μ | σ | Min | Max | Range |
| 1 | 0.061 | 0.013 | 0.038 | 0.087 | 0.048 | 0.148 | 0.041 | 0.085 | 0.241 | 0.157 |
| 2 | 0.111 | 0.015 | 0.069 | 0.138 | 0.069 | 0.135 | 0.037 | 0.089 | 0.210 | 0.121 |
| 3 | 0.142 | 0.021 | 0.115 | 0.182 | 0.067 | 0.196 | 0.051 | 0.120 | 0.294 | 0.174 |
| 4 | 0.201 | 0.021 | 0.170 | 0.245 | 0.075 | 0.309 | 0.080 | 0.202 | 0.548 | 0.346 |
| 5 | 0.302 | 0.025 | 0.246 | 0.337 | 0.091 | 0.568 | 0.076 | 0.418 | 0.694 | 0.277 |
| 6 | 0.374 | 0.031 | 0.318 | 0.426 | 0.107 | 0.805 | 0.146 | 0.557 | 1.052 | 0.495 |
| 7 | 0.484 | 0.144 | 0.198 | 0.665 | 0.467 | 0.576 | 0.069 | 0.409 | 0.666 | 0.257 |
| 8 | 0.730 | 0.108 | 0.426 | 0.876 | 0.450 | 1.041 | 0.283 | 0.602 | 1.765 | 1.163 |

μ = overall mean for the respective flight or in-field inspection, including all experimental plots. σ = standard deviations of the measured values. Min = minimum values observed in the flight or in-field inspection. Max = maximum values observed in the flight or in-field inspection. Range = Min–Max.

Similarly, Table 4 presents the mean values of $H_{DSM}$ and $H_{REF}$ for each treatment according to each flight. According to the analysis of variance (ANOVA), no significant differences were found among the applied treatments for $H_{DSM}$. On the other hand, significant differences among the treatments were only observed for the $H_{REF}$ values of the last two flights; that is, Flights 7 and 8.

**Table 4.** Mean values of $H_{DSM}$ and $H_{REF}$ according to flight inspection and treatments.

| Flight | $H_{REF}$ (m) | | | | | | $H_{DSM}$ (m) [ns] | | | | | |
|---|---|---|---|---|---|---|---|---|---|---|---|---|
| | T1 | T2 | T3 | T4 | T5 | $\mu_F$ | T1 | T2 | T3 | T4 | T5 | $\mu_F$ |
| 1 | 0.059 | 0.066 | 0.062 | 0.062 | 0.057 | 0.061 | 0.135 | 0.126 | 0.161 | 0.175 | 0.144 | 0.148 |
| 2 | 0.103 | 0.099 | 0.122 | 0.113 | 0.117 | 0.111 | 0.130 | 0.120 | 0.137 | 0.152 | 0.135 | 0.135 |
| 3 | 0.133 | 0.150 | 0.142 | 0.144 | 0.139 | 0.142 | 0.182 | 0.184 | 0.197 | 0.216 | 0.198 | 0.195 |
| 4 | 0.196 | 0.204 | 0.204 | 0.198 | 0.200 | 0.201 | 0.296 | 0.314 | 0.293 | 0.357 | 0.286 | 0.309 |
| 5 | 0.298 | 0.317 | 0.298 | 0.287 | 0.309 | 0.302 | 0.569 | 0.576 | 0.546 | 0.541 | 0.610 | 0.568 |
| 6 | 0.363 | 0.377 | 0.383 | 0.364 | 0.382 | 0.374 | 0.742 | 0.776 | 0.794 | 0.845 | 0.867 | 0.805 |
| 7 | 0.450 [bc] | 0.624 [a] | 0.340 [c] | 0.456 [bc] | 0.548 [ab] | 0.484 | 0.557 | 0.625 | 0.553 | 0.586 | 0.556 | 0.576 |
| 8 | 0.762 [ab] | 0.819 [a] | 0.668 [b] | 0.685 [b] | 0.719 [ab] | 0.730 | 0.982 | 1.223 | 0.863 | 1.109 | 1.025 | 1.044 |
| $\mu_T$ | 0.295 | 0.332 | 0.277 | 0.289 | 0.309 | | 0.449 | 0.493 | 0.443 | 0.497 | 0.477 | |

T1—control, T2–T5: 25%, 50%, 75%, and 100% of treated slaughterhouse effluent, respectively. $\mu_F$ = mean height value for each respective flight in the case of $H_{DSM}$ or in-field inspection in the case of $H_{REF}$. $\mu_T$ = mean value according to each respective treatment. Different letters in the same row indicate significant differences among treatments and ns = no significant differences observed according to the analysis of variance with a significance level of 0.05.

Additionally, in general, the mean height for both methods shows a growing tendency over time, which can be seen along the columns of each treatment in Table 4. Plant height was evaluated according to the growth stage datasets presented in Table 5. Figure 3 displays the regression models for each growth stage dataset along with their descriptive statistics.

In general, models performed better when considering data collected up until and including the grain filling stage ($R^2 = 0.74$) and flowering stage ($R^2 = 0.85$), with the latter presenting better performance. In contrast, the model using data collected up until and including the booting stage presented the weakest performance for estimating plant height.

The regression models were evaluated using 70% of the datasets for modeling and 30% for verification, randomly selected from each dataset of the respective growth stage. The modeling subset was used to build the models, and the verification data was used to evaluate its performance. Figure 4 shows the model performance and the relationship between real and estimated heights.

**Table 5.** Mean values for dry biomass (t ha$^{-1}$) according to each treatment.

| Growth Stage | DAS | T1 | T2 | T3 | T4 | T5 |
|---|---|---|---|---|---|---|
| Tillering | 19 | 0.111 | 0.112 | 0.175 | 0.123 | 0.179 |
| Booting | 33 | 0.680 | 0.589 | 0.773 | 0.729 | 0.782 |
| Flowering | 47 | 3.400 | 3.739 | 3.902 | 3.809 | 3.970 |
| Grain filling | 62 | 5.238 | 4.715 | 4.715 | 5.066 | 5.213 |

T1—control, T2–T5: 25%, 50%, 75%, and 100% of treated slaughterhouse effluent, respectively. DAS = days after sowing. No significant differences were observed among the treatments for each growth stage, with a confidence level of 0.05.

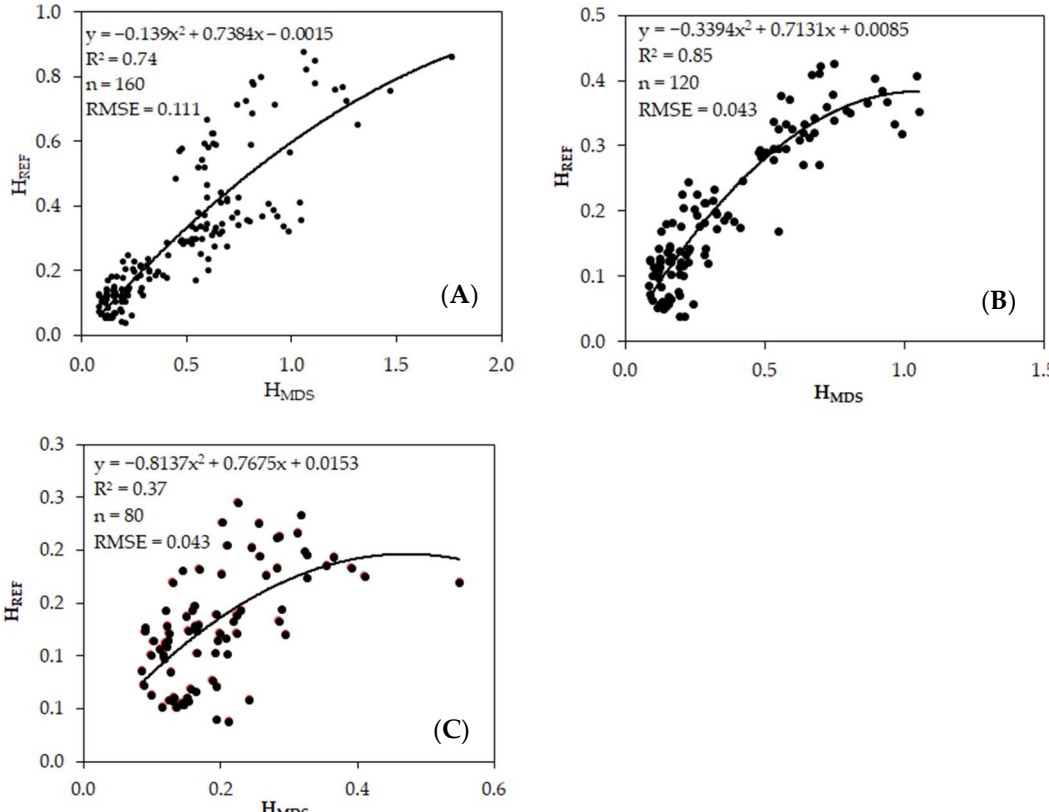

**Figure 3.** Regression models between H$_{REF}$ and H$_{DSM}$. (**A**) Regression model using all data; (**B**) regression model using data collected up until and including the flowering stage; (**C**) regression model using data collected up until and including booting stage. RMSE = Root Mean Square Error (m), n = number of observations. For all models $p < 0.05$.

Corroborating with the previous results (Figure 3), the model with the best performance was the one that used data collected up until and including the flowering stage, with an R$^2$ of 0.85 for the model (Figure 4C) and 0.85 for the estimate regression (Figure 4D). Using data collected up until and including the grain filling stage (all data) provided intermediate performance, R$^2$ was 0.73 and 0.78 for the model and estimate analysis (Figure 4A,B), respectively. Furthermore, models built with data collected up until and including the booting stage provided the weakest prediction performance (Figure 4E,F).

It is, therefore, understood that during the initial growth stages, the height of the plants estimated through aerial imaging does not represent the real height of the plants. On the other hand, later stages were demonstrated to accurately describe the real height of the plants with sufficient reliability [7,38].

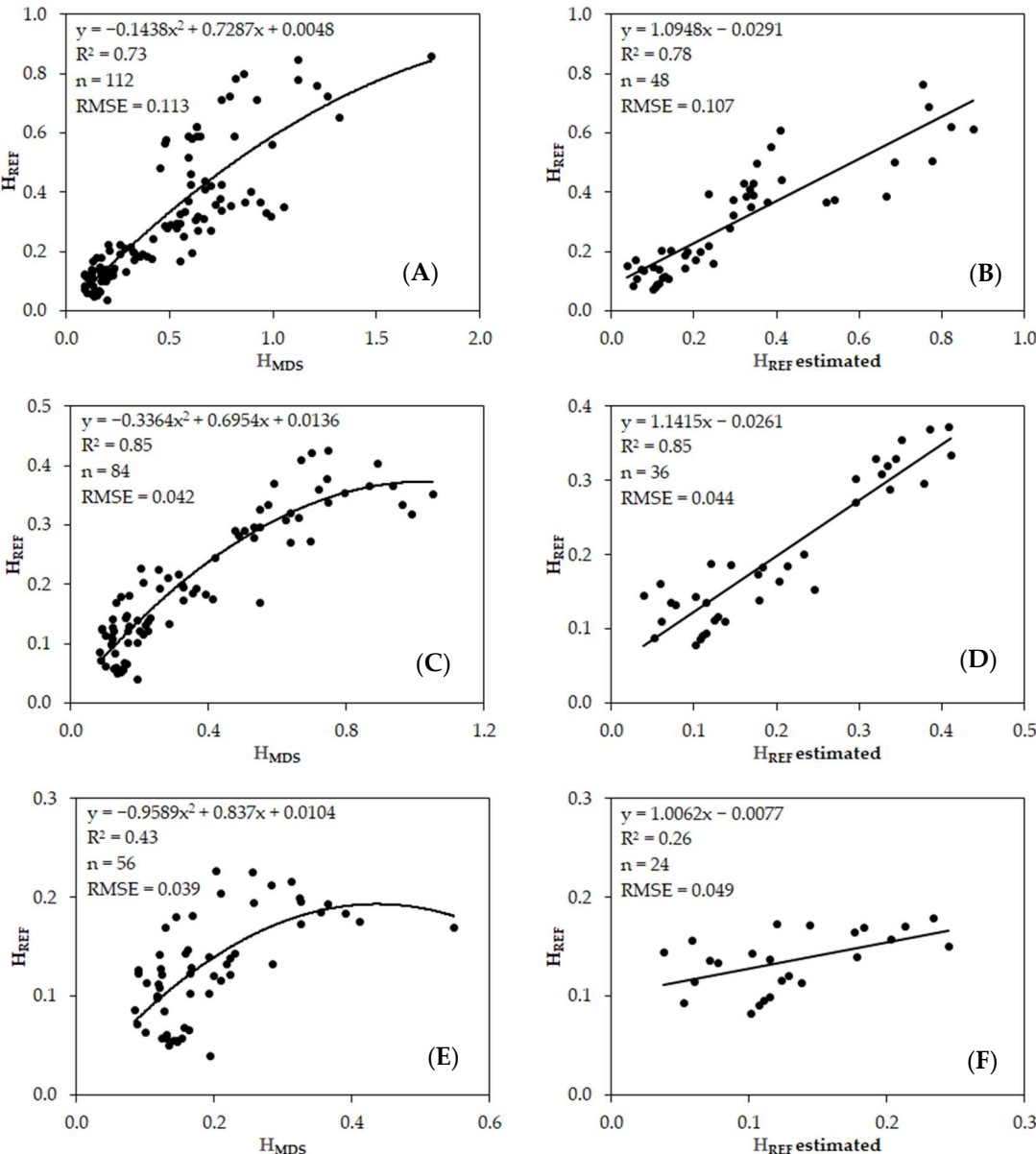

**Figure 4.** (**A**,**C**,**E**): Regression models between $H_{REF}$ and $H_{DSM}$ using modeling dataset for all stages, the flowering stage, and the booting stage respectively; (**B**,**D**,**F**): performance of the regression models with the validation dataset for all stages, the flowering stage, and the booting stage, respectively. RMSE = root mean square error (m), n = number of observations. For all models $p < 0.05$.

### 3.2. Biomass Assessment

Besides $H_{REF}$ and $H_{DSM}$, plant dry biomass (DB) was also obtained for each growth stage. The overall mean values of DB for each treatment are presented in Table 5. Descriptive statistics regarding the overall means of each growth stage are shown in Table 6, considering $H_{REF}$, $H_{DSM}$, and DB. It is possible to identify a growing tendency over time for all parameters evaluated, indicating that the expected behavior of the plants is being observed.

According to the analysis of variance (ANOVA) performed with data from each growth stage, no significant differences were observed among the treatments, with a confidence level of 0.05.

Similarly to the plant height assessment, data were split into subsets according to the growth stage for DB prediction modeling. The models were evaluated using 70% of the

data for modeling and 30% for verification. The modeling subset was used to build the models, and the verification data was used to evaluate its performance.

**Table 6.** Descriptive statistics for overall mean values for all parameters evaluated in the study.

| Flight | DAS | Growth Stage | Parameters | μ | σ | Median | Range |
|---|---|---|---|---|---|---|---|
| 1 and 2 | 19 | Tillering | $H_{REF}$ | 0.111 | 0.015 | 0.069 | 0.068 |
| | | | $H_{DSM}$ | 0.135 | 0.037 | 0.120 | 0.121 |
| | | | DB | 0.140 | 0.066 | 0.114 | 0.235 |
| 3 and 4 | 33 | Booting | $H_{REF}$ | 0.201 | 0.021 | 0.197 | 0.075 |
| | | | $H_{DSM}$ | 0.309 | 0.080 | 0.299 | 0.346 |
| | | | DB | 0.711 | 0.264 | 0.620 | 0.856 |
| 5 and 6 | 47 | Flowering | $H_{REF}$ | 0.374 | 0.031 | 0.369 | 0.107 |
| | | | $H_{DSM}$ | 0.805 | 0.146 | 0.770 | 0.495 |
| | | | DB | 3.764 | 0.638 | 3.755 | 2.442 |
| 7 and 8 | 62 | Grain filling | $H_{REF}$ | 0.730 | 0.108 | 0.756 | 0.450 |
| | | | $H_{DSM}$ | 1.041 | 0.283 | 1.026 | 1.163 |
| | | | DB | 5.136 | 0.918 | 5.031 | 3.434 |

DAS = days after sowing. μ = overall mean values for 20 plots in each flight mission for in-field inspection. σ = standard deviation. $H_{REF}$ = plant height measure through in-field inspection (m). $H_{DSM}$ = plant height derived from digital surface model (m). DB = dry biomass (g kg$^{-1}$).

Once DB had been determined through destructive analysis at the end of each growth stage, the respective last flight data for each of these stages were used in the analysis, indicated by the DAS values in Table 6.

The mean values of DB, in addition to $H_{REF}$ and $H_{DSM}$, show an increase in standard deviation over time, rising from 0.066 to 0.918. A similar pattern can be observed for all parameters studied.

For predicting biomass, three approaches were investigated. Firstly, a regression model was developed using $H_{DSM}$; secondly, $VI_{RGB}$s were used to develop a regression model to estimate dry biomass. Then, a combination of $H_{DSM}$ and $VI_{RGB}$ was evaluated to increase the model accuracy through multiple regression.

To investigate the most suitable index from Table 1 for predicting biomass, a Pearson's correlation matrix was constructed using $VI_{RGB}$ from the literature (Table 1) with scores considered strong (≥0.7), moderate (<0.7 and ≥0.5), or weak (<0.5). Only those with the greatest correlation were considered for the modeling.

Analyzing the matrix (Table 7), all of the $VI_{RGB}$s presented a strong correlation with DB and $H_{DSM}$. For DB, the TCVI showed the highest correlation score (0.7987), with an inversely proportional behavior. Likewise, VARI showed a strong correlation with $H_{DSM}$ (0.7880).

**Table 7.** Pearson's correlation matrix.

| | DB | $H_{DSM}$ | $H_{REF}$ | MPRI | NExG | VARI | GRRI | TCVI |
|---|---|---|---|---|---|---|---|---|
| DB | 1 | | | | | | | |
| $H_{DSM}$ | 0.8898 | 1 | | | | | | |
| $H_{REF}$ | 0.9249 | 0.8711 | 1 | | | | | |
| MPRI | 0.7797 | 0.7244 | 0.7059 | 1 | | | | |
| NExG | 0.7655 | 0.7151 | 0.6834 | 0.9935 | 1 | | | |
| VARI | 0.7880 | 0.7258 | 0.7141 | 0.9986 | 0.9891 | 1 | | |
| GRRI | 0.7912 | 0.7206 | 0.7109 | 0.9953 | 0.9880 | 0.9982 | 1 | |
| TCVI | −0.7987 | −0.7238 | −0.7371 | −0.9662 | −0.9378 | −0.9764 | −0.9761 | 1 |

DB = dry biomass. $H_{DSM}$ = height derived from images. $H_{REF}$ = in-field height measurement. MPRI = modified photochemical vegetation index. NExG = normalized extra green index. VARI = visual atmospheric resistance index. GRRI = green–red ratio index. TCVI = true color vegetation index.

Regression models were derived from the datasets regarding each growth stage, and the significance of the models was evaluated through regression analysis. For each growth stage subset, DB was modeled as a function of $H_{DMS}$, a function of TCVI, and as a combination of $H_{DSM}$ and TCVI. The best model for each growth stage along with descriptive statistics are presented in Table 8.

**Table 8.** Descriptive statistics for overall mean values.

| Growth Stage | Regression | Equation | n | $R^2$ | RMSE |
|---|---|---|---|---|---|
| Booting | DB vs. $H_{DSM}$ | $y = -8.9434x^2 + 6.5081x - 0.4773$ | 40 | 0.46 | 0.251 |
| | DB vs. TCVI | $y = -1.6049x + 1.1915$ | 40 | 0.80 | 0.153 |
| | DB vs. $H_{DSM}$ + TCVI | $y = -2.9654x_1 - 3.0222x_2 + 6.3065x_1x_2 + 1.9293$ | 40 | 0.84 | 0.137 |
| Flowering | DB vs. $H_{DSM}$ | $y = 4.9664x - 0.5290$ | 60 | 0.82 | 0.685 |
| | DB vs. TCVI | $y = -4.9779x + 3.2732$ | 60 | 0.60 | 1.031 |
| | DB vs. $H_{DSM}$ + TCVI | $y = 3.9959x_1 - 1.5556x_2 + 0.4172$ | 60 | 0.85 | 0.629 |
| Grain Filling | DB vs. $H_{DSM}$ | $y = -2.7730x^2 + 8.6046x - 1.1367$ | 80 | 0.84 | 0.859 |
| | DB vs. TCVI | $y = -6.6919x + 4.3162$ | 80 | 0.64 | 1.296 |
| | DB vs. $H_{DSM}$ + TCVI | $y = 3.4617x_1^2 + 7.0242x_1 - 3.5630x_2$ | 80 | 0.86 | 0.816 |
| Validation model | DB vs. $H_{DSM}$ + TCVI | $y = -1.7206x_1^2 - 1.8181 + 6.3040x_1 + 1.1599$ | 56 | 0.87 | 0.769 |

Significant $R^2$ and coefficients for all models ($p < 0.05$). The growth stage indicates which stages were considered in each model. The validation model is the model built with the modeling dataset (70% of available data). $x_1 = H_{DSM}$ and $x_2 = $ TCVI. DB = dry biomass (g kg$^{-1}$).

It is possible to observe that models developed with a combination of plant height and vegetation indices provided better estimates for plant biomass, with very close coefficients of regressions among the growth stages, producing values of 0.84, 0.85, and 0.86 for the datasets considering data collected up until the booting, flowering, and grain filling stages, respectively.

The model using data collected up until and including the booting stage and the $H_{DSM}$ was the weakest for estimating biomass. However, when considering the data collected up until and including the booting stage and the TCVI, the estimated performance of the model increased. For the subsets of all data and data collected up until and including flowering stage, the models using TCVI had the lowest $R^2$ values.

Overall, models with combinatory factors for $H_{DSM}$ and TCVI performed better for biomass estimation. TCVI alone proved to be good for estimating biomass in the initial stages of development, while for later stages, it might not be recommended. In contrast, plant height alone demonstrated better performance for estimating biomass during the later stages of black oat development.

The combinatory model for data collected up until and including the booting stage yielded an $R^2$ of 0.84, and the data collected up until and including the flowering stage produced an $R^2$ value equal to 0.85. Finally, the model built with all data (collected up until and including the grain filling stage) produced an $R^2$ value equal to 0.86. Although all of the models presented very close $R^2$ values, the model built using all data and a combination of $H_{DSM}$ and TCVI performed slightly better. This result corroborates with the findings from the literature, where combining plant height with vegetation index led to the acquisition of better estimative models.

Using all of the data, two subsets were obtained. Of the data, 70% was used to derive the model and 30% was used for validation. The validation subset was used in the validation model from Table 8. The predicted data were plotted against the real DB values obtained in the laboratory, and the relationship between the estimated and real DB values are presented in Figure 5. The validation of the regression model combining plant height and $VI_{RGB}$ provided an $R^2$ value equal to 0.82, thus demonstrating suitability for estimating biomass.

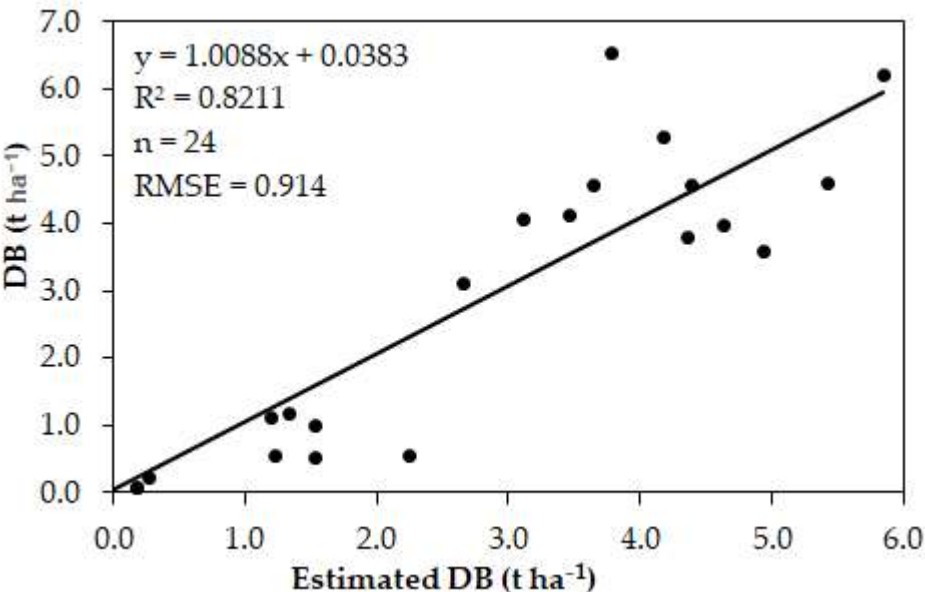

**Figure 5.** Validation for the model using combinatory regression with $H_{DSM}$ and TCVI.

## 4. Discussion

In this work, a new workflow for image acquisition and processing, without the use of ground control points and image pre-processing, was presented. The proposed methodology contains the image acquisition step (flight missions), the processing step with SfM software, and the last step, which includes extracting information for plant height determination and vegetation indices calculation.

The methodology workflow here presented can be summarized in Figure 1 and has demonstrated suitability for the proposed objective; that is, obtaining high-quality DSMs and orthoimages from UAV-based RGB images via a direct-to-process method, without the use of ground control points or image pre-processing, resulting in a faster and more accurate methodology for generating these photogrammetric products, and allowing for application regression models based on $H_{DSM}$ and $VI_{RGB}$, as well as the combination of $H_{DSM}$ and $VI_{RGB}$ successfully, which is discussed in this section.

The calculations of plant height and vegetation indices were performed using the algorithm developed by the authors, which was able to successfully read and extract information from the digital numbers of high-quality images. This information could correspond to the intensities of the R, G, and B channels at the pixel level in the orthoimages or the respective elevation value in the DSM. With the extracted information, plant height could be determined, and vegetation indices calculated.

Even though GCPs were not applied in the study, high-quality photogrammetric products were obtained, which have demonstrated suitability for estimating plant height and biomass. This result corroborates the findings of Yue et al. [37], indicating that for applications where the main interest is to estimate biomass, not using GCPs might have advantages related to a faster and less laborious workflow, without sacrificing the quality of estimates [26,28]. This is possible since applications that are interested in determining plant height or biomass through plant height and vegetation indices do not require high precision in the coordinates of the pixels on the image. This fact permits a faster analysis since the operator can go to the field, perform the flight mission, and start the processing, without the previously required extra in-field work, such as placing marks and obtaining high-precision reference points.

The average DSM resolution was 0.0062 mm pix$^{-1}$ and the orthoimage average resolution was 0.0124 m pix$^{-1}$ for flights at 20 m from the ground level. Table 2 presents the resolution information in detail. Moreover, the flight distance from the ground (20 m) proved adequate for assessing the black oat crop under the conditions in the study. From

the literature, a similar flight configuration at 25 m above ground level with a 12-MP camera obtained an average ground sampling distance of 0.0174 m pix$^{-1}$ [7].

Moreover, Bendig et al. [24] obtained images using a UAV coupled with a 16-MP RGB camera performing flights at 50 m from ground level and derived DSM images with 0.01 m resolution. Despite the fact that both of these studies used GCPs for increasing the precision of coordinates, the present study was able to produce comparable photogrammetric products with the advantage of reduced field work, leading to a faster workflow with lower computational costs and eliminating the need for expensive high-precision equipment.

The $H_{DSM}$ and $H_{REF}$ values presented similar behavior in the sense that both of them showed a growing pattern, indicating that the growth of the crop was observed. $H_{REF}$ varied from 0.061 to 0.730 m and $H_{DSM}$ ranged from 0.148 to 1.041 m. Comparable values are found in the literature for black oats, where $H_{REF}$ ranged from 0.120 to 0.650 m and $H_{DSM}$ from 0.08 to 0.81 m [7]. For other crops, such as barley and sorghum, estimated plant heights usually have lower values than the reference heights, which was not observed in this study. Regardless, $H_{DSM}$ can describe the growth behavior of plants [7,11,24], and the results of this study reassert its suitability for modelling plant growth.

Moreover, $H_{DSM}$ presented mean values up to two times greater than the mean values observed for $H_{REF}$. This behavior is not observed in most studies that use GCPs for correcting coordinates, including altitude [7,11,38,39]. However, the growing pattern is respected, a fact that corroborates the good prediction results found in the present study. Regarding the range of plant heights, both $H_{REF}$ and $H_{DSM}$ have shown greater data dispersion for later growth stages, similar to the results found in the literature for black oats and other crops [7,27]. This statement is corroborated by the increased standard deviation observed during Flights 6, 7, and 8. In general, $H_{DSM}$ had a greater standard deviation for all growth stages compared to $H_{REF}$.

This could be explained by the natural characteristics of a DSM; since it builds a surface model over all the points existing in the image, it consequently includes points that do not belong to any plant, such as rows or gaps in the field, making the $H_{DSM}$ variability greater. This is less evident in the early development stages since the plant heights are closer to the soil, where the points from rows and gaps are. When it comes to later growth stages, this difference becomes much higher; therefore, the variability increases.

Plants are measured considering their heights in their natural position. At the end of the flowering and at the beginning of grain filling stages, grains are being formed, generating a greater weight at the extremities of the plants, which makes the stems tilt down and, consequently, the heights of some plants might be lower. As the grain filling was not homogeneous during the last stages, plant heights presented greater variation. Therefore, it can be said that for modeling the plant height and biomass of black oats, the later growth stages, starting at approximately 47 DAS, which include the final days of flowering and the entire grain filling stage, might not be adequate. The same results have been found in the literature [7,28].

Moreover, the treatments applied to the crop demonstrated no influence on plant height during the initial growth stages. $H_{REF}$ showed significant differences among treatments only during the last growth stage. This fact might be explained by the accumulation of nitrogen during the experiment, reaching levels sufficient for influencing growth at the final development stage. No significant differences were observed for $H_{DSM}$. Likewise, the different dosages of STE did not influence dry biomass, which has also shown an increasing pattern over time, similar for all treatments.

The mean values of DB observed in this study were 0.140, 0.711, 3.764, and 5.136 t ha$^{-1}$ for the tillering, booting, flowering, and grain filling growth stages, respectively. In contrast, Luz et al. [40] studied the influence of different dosages of nitrogen applied through irrigation and found significant differences by destructive foliar analysis and lower mean dry biomass values, ranging from 0.683 to 1.832 t ha$^{-1}$. On the other hand, mean values ranging from 0.690 to 8.150 t ha$^{-1}$ were observed in a different study [7]. As studies

concerning the influence of nitrogen on black oat are scarce, further investigation is needed to understand these behaviors.

The methodology has shown suitability for estimating real plant height based on $H_{DSM}$ (Figure 1). It was demonstrated that collecting data from sowing during the end of the flowering stage is better for estimating plant height since the regression model for this growth interval presented an $R^2$ of 0.85 and an RMSE of 0.043. On the other hand, using a single stage for estimating plant height has proven to be inadequate, which has also been observed by Jannoura et al. [41].

Being aware of this behavior is important, because it shows that estimating plant height up until approximately 33 DAS, during the booting stage, might not reflect the real growth status of the plants. In contrast, the tests showed that it is not necessary to wait until the last stage of development to have a more accurate estimate. The model validation tests corroborate these statements, since the best performance for estimating plant height was observed when the data collected during the flowering stage was used (Figure 2). The linear relationship between estimated $H_{REF}$ and actual $H_{REF}$ yielded an $R^2$ value equal to 0.85. These values are comparable to those found in other studies concerning black oats [7].

When using all the data available—that is, data collected at the end of the grain filling stage—the model presented a moderate-to-good response for estimating plant height ($R^2 = 0.73$ and RSME 0.113), and the relationship between the estimated and actual height values presented an $R^2$ value of 0.78. Thus, in situations where UAV availability is restricted, we recommend that flights be performed during the initial-to-intermediate growth stages, since results using data collected during the booting stage performed better than single-stage data.

Another point of discussion regarding the methodology presented in this study is the fact that a zero ground height reference was determined for each flight, keeping reference areas clean, while other studies have used the first flight as the zero reference [24,42,43], which brings bias to the data as during the first flight, plants might have already emerged [24,31].

To evaluate the correlation among the response variables ($H_{DSM}$, $VI_{RGB}$, and DB), a Pearson's correlation matrix was used. In this study, we considered the correlation scores to be either high ($\geq 0.7$), intermediate ($\geq 5$ and $<7$), or low ($<5$). As shown in the literature, plant DB has a strong correlation with plant height, which was also observed in this study [7,9,11,27,30,31]. The $H_{DSM}$ showed a correlation score of 0.89, considered high. Moreover, all of the $VI_{RGB}$s demonstrated a strong correlation with DB, with the best performance presented by TCVI ($-0.79$). $VI_{RGB}$ vegetation indices have also been tested for other crops, such as peas and oats, and have demonstrated a strong correlation with plant biomass [31] Therefore, $H_{DSM}$ and TCVI were employed for modeling dry biomass.

Vegetation indices previously applied to corn, grass, wheat, and rice in the literature have demonstrated a strong correlation with black oat physiological behavior; thus, they could be used for biomass estimations. $H_{DSM}$ and TCVI were used as factors for building the DB estimation models. Modeling DB using $H_{DSM}$ with data collected up until and including the booting stage resulted in a second-order polynomial equation, but, even though the model was significant ($p < 0.05$), its $R^2$ of 0.46 was considered low and demonstrated weak potential for estimating DB with good accuracy.

When using data collected until the end of the flowering stage, the fitted model had a linear behavior and performed better for estimating DB, with an $R^2$ value equal to 0.82. Lastly, using data collected up until and including the grain filling stage, the best model was a second-order polynomial regression, which had the higher performance for estimating DB by $H_{DSM}$, presenting an $R^2$ value of 0.84. Acorsi et al. [7] found that prediction models performed better when using data from the flowering stage ($R^2 = 0.94$).

The suitability of $VI_{RGB}$ for estimating biomass has been observed in the literature [9,18,28], but in general, the quality of the models has been lower than what was found in this study. Considering TCVI alone for estimating biomass, the regression models showed a linear behavior for the subsets of all growth stages with the best performance when using data collected up until and including the booting stage ($R^2 = 0.80$). However,

for the later growth stages, the linear regression models presented weaker coefficients of determination (0.60 and 0.64 for the flowering and grain filling stages, respectively). These results indicate that, for an evaluation of biomass needed to be taken at the initial stages of black oats development, the use of TCVI is recommended.

Finally, $H_{DSM}$ and TCVI were combined to produce multiple regression models. When combining the factors, models presented a higher $R^2$, thus proving them to be better at estimating dry biomass. This result reaffirms other studies that have observed improvement in dry biomass modeling for grassland and herbaceous crops by combining $H_{DSM}$ and $VI_{RGB}$ [44,45]. For all the growth stages, second-order polynomial regressions had the best performance (Table 7), with $R^2$ values of 0.84, 0.85, and 0.86 for the data collected up until and including the booting, flowering, and grain filling stages, respectively. Although the $R^2$ values were very close to each other, the model using the data collected up until and including the grain filling stage performed better.

Therefore, the dataset with data collected up until and including the grain filling stage was used for the validation of the second-order polynomial regression model. The validation model using 70% of the dataset is presented in Table 7, and the linear relationship between actual and estimated DB values using 30% of the dataset is shown in Figure 4. The validation model yielded an $R^2$ value of 0.87, while the linear regression had an $R^2$ value of 0.82. These results agree with the findings in the literature regarding the combination of $VI_{RGB}$ and $H_{DSM}$ [24].

In spite of the observed plant height estimative, for estimating black oat DB, this study recommends using data from all growth stages, from sowing through the grain filling stage, and a combination of $H_{DSM}$ and TCVI. Furthermore, it is important to highlight that the presented study did not use GCPs for correcting the coordinates of the DSMs and orthoimages, presenting a methodology that permits performing the inspection flights, generating photogrammetric products using commercial software, extracting the information of interest (i.e., plant height and TCVI), using an open-source algorithm written in Python, and, finally, using those inputs to estimate DB production of black oats with high accuracy and trustworthiness.

Even though it was not the focus of this study, as black oat has the potential for use as green manure and animal feed, further investigation is necessary to understand the dynamics of nitrogen and other foliar nutrients under the conditions presented here, so it can be used as an input for building better models and estimating the nutritional status of the green manure. Moreover, it is necessary to proceed with research considering the bromatological content of the black oat, so models can be developed and correlated with animal nutrition aspects.

## 5. Conclusions

High-quality DSMs and orthoimages were obtained from UAV-based RGB images and applied to estimating black oat height and biomass. Elevation information extracted from the DSMs was first used to determine plant height along the growth stages of the plants. The coefficient of determination ($R^2$) among $H_{DSM}$ and $H_{REF}$ was found to be intermediate to high with the best results when using data collected up until and including the flowering stage ($R^2 = 0.85$), while using data collected until the end of the grain filling stage yielded an $R^2 = 0.73$. When using data collected up until and including booting stage, the regression showed the lowest performance ($R^2 = 0.43$), indicating that data collected during the initial stages of development might not represent the real height of the plants. It is thus recommended that data collected up until and including flowering be used.

From the studied $VI_{RGBs}$, TCVI has presented the strongest correlation with dry biomass and also a high correlation with $H_{DSM}$ according to Pearson's correlation matrix (Table 7).

Dry biomass was studied through regression models. For all growth stages, models that combined $VI_{RGB}$ and $H_{DSM}$ showed the best performance for predicting the dry biomass of black oat, with $R^2$ values of 0.84, 0.85, and 0.86 when using data collected

until the end of the booting, flowering, and grain filling stages, respectively. As these $R^2$ values are very close, it can be said that a good estimation of productivity, through dry biomass, can be made at the initial growth stages of black oat. When analyzing DB with a single factor, it is recommended that $VI_{RGB}$ be used when only data collected up until and including the booting stage is available, but for the later growth stages, modeling dry biomass based on $H_{DSM}$ proved to be better.

Therefore, the methodology demonstrated potential for obtaining photogrammetric products via a faster and direct-to-process method, resulting in high-quality DSMs and orthoimages. Without using GCPs, the data extracted from these DSMs and orthoimages provided good-quality models for estimating plant height and dry biomass. Further studies need to be performed to test the models with different sets of data from black oats and other grasses to improve estimation capability.

**Author Contributions:** Conceptualization, L.R.T. and F.R.; methodology, L.R.T. and F.R.; software, L.R.T.; validation, F.R., T.M.G. and L.B.; formal analysis, F.R. and L.R.T.; investigation, L.R.T. and L.B.; data curation, L.R.T.; writing—original draft preparation, L.R.T.; writing—review and editing, L.R.T., F.R., L.B. and T.M.G.; supervision, F.R.; project administration, F.R.; funding acquisition, F.R. and T.M.G. All authors have read and agreed to the published version of the manuscript.

**Funding:** This research was funded by Coordenação de Aperfeiçoamento de Pessoal de Nível Superior (CAPES), grant number 88882.378441/2019-01.

**Data Availability Statement:** Publicly available datasets were analyzed in this study. The data will be made available posteriorly to this publication. The data will be found here: [https://repositorio.usp.br/] under the title of this research paper. Accessed on 17 February 2023.

**Acknowledgments:** The authors would like to thank the University of São Paulo Faculty of Food Engineering and Animals Science and "Luiz de Queiroz" College of Agriculture along with their staff. The authors also acknowledge the funding from Coordenação de Aperfeiçoamento de Pessoal de Nível Superior (CAPES).

**Conflicts of Interest:** The authors declare no conflict of interest.

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
