# Peer review of "Estimating Black Oat Biomass Using Digital Surface Models and a Vegetation Index Derived from RGB-Based Aerial Images"

_remotesensing, doi:10.3390/rs15051363_

Round 1

Reviewer 1 Report

This work propose to use UAV-based RGB images in a direct-to-process way (without the use of ground control points or image pre-processing) to estimate plant height  and dry biomass (DB), the latter based on plant height and RGB vegetation indices, applying the methodology in black oat cultivation

The paper presents a good bibliographic review, with some high impact citations.

However, the manuscript is not yet ready for publication, but it is very promising if the authors are willing to make the following small minor revisions

-       Missing units in most tables

-       Lines 173-174: It is said “Data were analyzed according to the treatments to identify significant differences due to nitrogen dosages through analysis of variance (ANOVA)” but then these results are not displayed.

-       Figure 2 is not explained in the text, the only explanation given is in the title of the figure.

-       Line 255 indicates table 3, but corresponds to table 5.

-       Lines 259-260 states “data were split into subsets according to the growth stage for DB prediction modeling”. More detail should be given on how this has been done, the proportion of cultivation still growing...

-       Lines 276-277 refers to figure 6, when in fact it is table 6.

-       Line 298 “persons’ correlation matrix”: error

-       The figure shown at the end " Graphical abstract " should be incorporated in the text

Author Response

Dear Reviewer,

Please find below, in this document, the responses to the questions, comments and suggestions made.

I kindly made sure to answer all points stated and to incorporate all suggestions into the text so our manuscript could be improved and enriched. All changes to the manuscript were made using the track changes option to make it clear and easy to find.

I highly appreciate your attention to our manuscript, If you have any further questions please let me know.

Answers to Reviewer:

-       Missing units in most tables

Following this comment, all tables were double-checked and any missing units were added to the table. The values might appear in the title, the body, or the footnote of the respective table. This was done to make sure all units are presented and the reading is fluid. Also, corresponding units within the body of the article were double-checked to guarantee all units were provided correctly.

-       Lines 173-174: It is said “Data were analyzed according to the treatments to identify significant differences due to nitrogen dosages through analysis of variance (ANOVA)” but then these results are not displayed.

The ANOVA test was performed for the plant height values obtained by in-field (HREF) and aerial imaging processing (HDSM) as well as for the biomass values.

For plant height, the results were initially presented by the superscript lowercase letters in Table 4 and the description was in the table’s footnote. Following the reviewer’s comment and intending to make it clear in the main text, the results are now described in the paragraph right before Table 4 (lines 262-266). It is important to mention that lowercase letters were only used for the flights in which significant differences were observed among the treatments, in our case, only in flights 7 and 8 for the reference height (HREF). For heights obtained by aerial images (HMDS), no significant differences were observed among the treatments for any flight, so we used the symbol ns (non-significant) to indicate that no significant differences were observed. This information is now presented in the footnote of Table 4 with more detail. Also, we corrected the description in the footnote to make it clear and provide more information to the reader.

For the biomass values, the results of the ANOVA were included in the paragraph right before Table 5 (lines 302-307). The information was also added to the Table’s footnotes. In this case, no significant differences were observed.

-       Figure 2 is not explained in the text, the only explanation given is in the title of the figure.

The description of Figure 2 is presented in the last two paragraphs right before the figure itself is presented (lines 284-293). Following the reviewer’s comment, adjustments to the text have been made to make it clear.

-       Line 255 indicates table 3, but corresponds to table 5.

This correction has been made in the manuscript.

-       Lines 259-260 states “data were split into subsets according to the growth stage for DB prediction modeling”. More detail should be given on how this has been done, the proportion of cultivation still growing...

To make it clear for the reader and better state the procedures we’ve followed, we have included more information about how the data were split at the end of the material and methods section, in the last two paragraphs. We also included more descriptions in the paragraph that follows Table 5.

We have three datasets considering the growth stages. The first dataset includes data from sowing until the grain filling stage, that is, all data available. The second dataset contains data from sowing until the flowering stage and lastly, the third dataset includes data from sowing until the booting stage. For each of these three stages, we separately and randomly selected 70% of the data for deriving models and 30% for testing its performance.

-       Lines 276-277 refers to figure 6, when in fact it is table 6.

This correction has been done to the text. Also, all referencing of tables and figures was double-checked to avoid the same mistake.

-       Line 298 “persons’ correlation matrix”: error

Correction made in the text.

-       The figure shown at the end " Graphical abstract " should be incorporated into the text

A paragraph at the beginning of the material and methods section was added to incorporate the graphical abstract in the main text.

Reviewer 2 Report

The manuscript is devoted to the development of algorithms for estimating the biomass of black oats. The scientific work is quite relevant, which is shown by the authors in the Introduction. The study is based on the results of [9, 18-21] and is their application to the biomass of black oats.

There are the following comments on the methodology: 

- it is first necessary to theoretically substantiate how vegetative indices (Table 2) for determining the biomass of corn, grass biomass, and nitrogen of rice, corn, and wheat can be associated with the determination of biomass (or plant height) of black oats,

- the issue of the influence of the solar radiation spectrum on the results of field measurements has not been sufficiently considered.  

The results obtained and their discussion is interesting, but the application of various (linear or quadratic) approximation equations and the relatively low (0.73-0.86) coefficient of determination raises questions.

All references given in the Manuscript are relevant to the study. For some reason, there is a duplicate numbering in the list of references.

Additional remarks:

1. The equation entry for TCVI (Table 1) is not clear: 255*0.4.

2. I recommend the authors write a brief summary conclusion (section Conclusion).

Author Response

Dear Reviewer,

Please find below, in this document, the responses to the questions, comments and suggestions made.

I kindly made sure to answer all points stated and to incorporate all suggestions into the text so our manuscript could be improved and enriched. All changes to the manuscript were made using the track changes option to make it clear and easy to find.

I highly appreciate your attention to our manuscript, If you have any further questions please let me know.

Answers to Reviewer:

Comments and Suggestions for Authors

There are the following comments on the methodology:

- it is first necessary to theoretically substantiate how vegetative indices (Table 2) for determining the biomass of corn, grass biomass, and nitrogen of rice, corn, and wheat can be associated with the determination of biomass (or plant height) of black oats

These aspects were first presented in the third paragraph of the introduction. However, following the reviewer’s comment, we understood the importance of including more information to better substantiate the relationship among corn, grass, rice and wheat and black oat regarding biomass and nitrogen determination and relationships between biomass and plant height (Lines 46 -62).

These new paragraphs show now that, from the literature, studies have demonstrated a strong correlation between biomass and nitrogen at the same time that biomass relates to plant height. Similarly, some studies corroborate the relationship between productivity (biomass) and nutritional status (nitrogen) through the use of vegetation indices.  Therefore, plant height and vegetation index can be used to estimate biomass. Moreover, the relationships already discussed and the similarity of black oats and the other major cereals from the grass family (grass, corn, wheat and rice, for instance) assure the possibility of applying the same vegetation index equations once used to study corn, wheat, grass and rice to assess black oat.

- the issue of the influence of the solar radiation spectrum on the results of field measurements has not been sufficiently considered. 

In this study, all the assessment has been made based on visible band images in the RGB color space. Considering the image acquisition method, since the sensor coupled on the unmanned aerial vehicle was not sensible to other bands of the solar spectrum, for instance, infrared and ultraviolet, we were able to only capture visible band information. To avoid changes in natural light radiation, all flights were performed at approximately noon on days of clear sky (Lines 72 to 82 in the introduction and 167-169 in the methods section).

In-field reference measurements were only performed for plant height using a ruler, thus not applying for solar radiation influence.

Following the reviewer’s consideration, we have included mre details that concern the solar radiation spectrum in the methods section of the article. Also, to enrich the discussion in the article, more details about the reflectance in the visible and non-visible spectrum were incorporated (Line 72)

-The results obtained and their discussion is interesting, but the application of various (linear or quadratic) approximation equations and the relatively low (0.73-0.86) coefficient of determination raises questions.

The data modeling was performed taking into consideration the best results for RMSE and R² of the models that presented p<0.05, that is, significant models.

Linear and polynomial regressions were considered with the significance of the coefficients always being analyzed and any non-significant parameter was dropped. This process was repeated for all data sets, so the best model for each case studied was found with the guarantee that all models were significant (p<0.05).

Polynomial regressions of order greater than two usually do not make sense for growth modeling, due to the natural behavior of the plants and the behavior of these models.

Also, in comparison with relevant references within the same field of study, the quality of the models we have found are comparable to the literature such as [7], which found R² ranging from 0.69 to 0.94 and [24], whose found R² of 0.82 for the regression between plant height and dry biomass, or even higher [9] (R² ranging from 0.56 to 0.88). Moreover, the combination of plant height and vegetation indices corroborates the founds in the literature [34, 35].

-All references given in the Manuscript are relevant to the study. For some reason, there is a duplicate numbering in the list of references.

Following the reviewer’s comment, we have double-checked the citations to avoid duplicates.

Additional remarks:

  1. The equation entry for TCVI (Table 1) is not clear: 255*0.4.

The equation has been corrected.

  1. I recommend the authors write a summary conclusion (section Conclusion).

Following the reviewer’s suggestion, a conclusion section has been added to the article.

Reviewer 3 Report

The manuscript titled “Estimating black oat biomass using digital surface models and vegetation index derived from RGB-based aerial images” estimated the plant height using the DSM, and then estimate the dry biomass of black oat with VI, based on the UAV RGB images. The results found that H and VI performed the best results for biomass estimation.

However, the innovation of this manuscript is weak. It is more like an experiment report of a remote sensing application, and it is not completed.

First, the original UAV images and the processed images (e.g. orthoimages and DSM images) can not be found in the manuscript, which should be indispensable and show in the sections of Experimental site and Results. The authors showed them in the Graphical abstract, so why not in the main portion of the manuscript?

Second, the authors declared in the Abstract and Discussion sections that “This study intended to obtain high-quality digital surface models (DSM) and orthoimages from UAV-based RGB images in a direct-to-process way, that is, without the use of ground control points or image pre-processing.” “In this work, a new approach for image acquisition and processing, without the use of ground control points and image pre-processing was presented”. However, in Line 141, “The orthoimages and DSMs were obtained using the SfM software Agisoft Metashape Professional (Version 1.6.4, Agisoft LLC, St. Petersburg, Russia) in its free trial standard version.” The authors did not develop a new program or module to process the UAV images.

Third, a Conclusion section is recommended.

All these issues made the results of this manuscript not complete and convincing.

In addition, there are many mistakes in the manuscript, to name a few.

Line 9, “black oats” should be “black oat”, considering the context.

Line 24, “IVRGB”.

Line 55, “notoriety” is not appropriate.

Line 99, “It is important to mention that for situations that require greater precision of coordinates, such as variable rate fertilizer applications, GCPs are indispensable.” Based on this sentence, why the authors did not use the GCP to obtain precise orthoimages?

Line 148, 170, 275, 277, 397, 402, “IVRGB”.

Line 148, “To obtain the HDSM and IVRGB the experimental plots for each flight were clipped from the original DSM and orthoimage”, it seems a comma is missing.

Line 168, “IVs”.

Line 161, “determine and store the mean height values for each reference plot”. How did the authors determine the mean height?

Line 299, Table 7, HDSM as x1, TCVI as x2? The authors should make it clear.

Line 453, the authors may want to add a section of Concluion here.

Author Response

Dear Reviewer,

Please find below, in this document, the responses to the questions, comments and suggestions made.

I kindly made sure to answer all points stated and to incorporate all suggestions into the text so our manuscript could be improved and enriched. All changes to the manuscript were made using the track changes option to make it clear and easy to find.

I highly appreciate the attention to our manuscript, if you have any further questions please let me know.

Answers to Reviewer:

The manuscript titled “Estimating black oat biomass using digital surface models and vegetation index derived from RGB-based aerial images” estimated the plant height using the DSM, and then estimate the dry biomass of black oat with VI, based on the UAV RGB images. The results found that H and VI performed the best results for biomass estimation.

However, the innovation of this manuscript is weak. It is more like an experiment report of a remote sensing application, and it is not completed.

First, the original UAV images and the processed images (e.g. orthoimages and DSM images) cannot be found in the manuscript, which should be indispensable and shown in the sections of Experimental site and Results. The authors showed them in the Graphical abstract, so why not in the main portion of the manuscript?

After reading the reviewer’s comment, the authors have understood the importance of including the referred images in the main text. Because of that, a new image was included (now, figure 2) containing the photogrammetric products from one of the flight missions performed to exemplify the obtained results.

Second, the authors declared in the Abstract and Discussion sections that “This study intended to obtain high-quality digital surface models (DSM) and orthoimages from UAV-based RGB images in a direct-to-process way, that is, without the use of ground control points or image pre-processing.” “In this work, a new approach for image acquisition and processing, without the use of ground control points and image pre-processing was presented”. However, in Line 141, “The orthoimages and DSMs were obtained using the SfM software Agisoft Metashape Professional (Version 1.6.4, Agisoft LLC, St. Petersburg, Russia) in its free trial standard version.” The authors did not develop a new program or module to process the UAV images.

What is proposed in this study is a methodology that does not require extra-field work for image processing and allows the obtaining of photogrammetric products in a faster way when compared to methodologies that use ground control points. That is, performing the flight and, right after, processing the images, without having to prepare the experimental field with references of known coordinates or perform pre-processing of the images.

The proposed approach contains the image acquisition step (flight missions with the UAV), processing using commercial software and later an algorithm for information extraction, such as the digital numbers of the images that could correspond to R, G and B channel intensities (in orthoimages) or elevation (DSM) and use that information to determine vegetation indices and plant height. Information regarding the algorithm's functioning is presented in the main text in the last three paragraphs of section 2.3.

Therefore, the new methodology that is proposed focused on the workflow from performing the flight mission to extracting information of interest and computing plant height and vegetation indices.

Intending to make the text more comprehensible, we have included more information following the reviewer’s comments. Lines 371-379.

Third, a Conclusion section is recommended.

Conclusion section was added to the manuscript following the reviewer’s comment.

All these issues made the results of this manuscript not complete and convincing. In addition, there are many mistakes in the manuscript, to name a few.

Line 9, “black oats” should be “black oat”, considering the context.

-Correction made

Line 24, “IVRGB”.

-Correction made at line 24 and throughout the entire manuscript

Line 55, “notoriety” is not appropriate.

-Correction made.

Line 99, “It is important to mention that for situations that require greater precision of coordinates, such as variable rate fertilizer applications, GCPs are indispensable.” Based on this sentence, why the authors did not use the GCP to obtain precise orthoimages?

The authors of the present study intended to estimate biomass based on plant height and vegetation indices, which is an application that does not require precise coordinates of orthoimages for high-quality results, since the precise location of the plants in the field would not influence its height or index value. It is important to mention that GCPs do not influence the resolution of the photogrammetric products. References that base this information are found in the manuscript (lines 117-126).

In contrast, applications such as autonomous machinery fertilization or agrochemical dosages in variable rates would be directly influenced by the exact location of the plants, that is, images with low precision in their coordinates could result in the wrong application of the inputs. Those are the applications where GCPs might be indispensable. This is not the case in this study, thus, not using GCP would bring the advantage of eliminating the need for in-field operations that would include going within the field to place reference points, requiring expensive equipment such as precise RTK systems, and demanding advanced knowledge labor, among others. Only after all this preparation is made it would be possible to perform the flight missions and the next steps in the workflow would have computational cost to pre-process the images for corrections.

Therefore, when GCPs are not used, the entire workflow is faster, requires less computational power and still gives reliable results, as seen in this research. The authors brought discussions comparing studies that used GCPs to show that, for the proposed application, it is possible to have a good methodology and good results without using GCPs.

More information has been added, considering the reviewer’s comments to enrich the manuscript (lines 389-393).

Line 148, 170, 275, 277, 397, 402, “IVRGB”.

Correction made

Line 148, “To obtain the HDSM and IVRGB the experimental plots for each flight were clipped from the original DSM and orthoimage”, it seems a comma is missing.

Correction made.

Line 168, “IVs”.

Correction made

Line 161, “determine and store the mean height values for each reference plot”. How did the authors determine the mean height?

Since the DSM contains information about elevation, that is, the pixels carry values that correspond to the height of that point on the image, the mean height values were determined for each plot by reading the value of each pixel, computing its sum and dividing by the total number of pixels

For reference height, 30 measurements with a ruler were made in each experimental plot for each date of flight. The average of the measurements was calculated, generating one mean value for each plot. This information has been made more evident in the manuscript (lines 193-190)

Line 299, Table 7, HDSM as x1, TCVI as x2? The authors should make it clear.

Following the reviewer’s comment, it was included in the footnote of the table to make clear how to read the variables in the equations. X1 and X2 were used to make the equations shorter and the reading more fluid.

Line 453, the authors may want to add a section of Concluion here.

A conclusion section has been included.

Round 2

Reviewer 3 Report

I am happy to see that the authors addressed all the coments.

Good luck to your further study.